# Three ParA Dimers Cooperatively Assemble on Type Ia Partition Promoters

**DOI:** 10.3390/genes12091345

**Published:** 2021-08-28

**Authors:** François Boudsocq, Maya Salhi, Sophie Barbe, Jean-Yves Bouet

**Affiliations:** 1Laboratoire de Microbiologie et Génétique Moléculaires, Centre de Biologie Intégrative (CBI), Centre National de la Recherche Scientifique (CNRS), Université de Toulouse, UPS, F-31062 Toulouse, France; maya.salhi-hernandez@inrae.fr (M.S.); jean-yves.bouet@univ-tlse3.fr (J.-Y.B.); 2CNRS, Toulouse Biotechnology Institute (TBI), Université de Toulouse, INRAE, INSA, F-31077 Toulouse, France; sbarbe@insa-toulouse.fr

**Keywords:** bacterial DNA segregation, partition promoter organization, ParA, plasmid F, winged-HTH

## Abstract

Accurate DNA segregation is essential for faithful inheritance of genetic material. In bacteria, this process is mainly ensured by partition systems composed of two proteins, ParA and ParB, and a centromere site. Auto-regulation of Par operon expression is important for efficient partitioning and is primarily mediated by ParA for type Ia plasmid partition systems. For the F-plasmid, four ParA_F_ monomers were proposed to bind to four repeated sequences in the promoter region. By contrast, using quantitative surface-plasmon-resonance, we showed that three ParA_F_ dimers bind to this region. We uncovered that one perfect inverted repeat (IR) motif, consisting of two hexamer sequences spaced by 28-bp, constitutes the primary ParA_F_ DNA binding site. A similar but degenerated motif overlaps the former. ParA_F_ binding to these motifs is well supported by biochemical and modeling analyses. Molecular dynamics simulations predict that the winged-HTH domain displays high flexibility, which may favor the cooperative ParA binding to the promoter. We propose that three ParA_F_ dimers bind cooperatively to overlapping motifs, thus covering the promoter region. A similar organization is found on closely related and distant plasmid partition systems, suggesting that such promoter organization for auto-regulated Par operons is widespread and may have evolved from a common ancestor.

## 1. Introduction

Genome stability depends on faithful segregation of replicated DNA to daughter cells. Stable inheritance of low-copy-number bacterial replicons (chromosomes and plasmids) requires an active partition mechanism. Each of such replicons typically carries a self-specific partition module comprising a two-gene operon and an adjacent set of short sequence repeats that acts as a centromere. The first gene encodes an ATPase and the second a centromere-binding protein. Partition systems have been classified into three main classes depending on the type of ATPase encoded with Type I, II and III, which are characterized by deviant Walker A, actin-like and tubulin-like NTPases, respectively (for review [1,2]). Type I systems, generically termed ParABS, are the most common type in the bacterial world, being widespread on low-copy-number plasmids and the only type present on chromosomes. ParA ATPases act on large nucleoprotein complexes formed by ParB on *parS* sites to drive the duplicated centromere DNA to opposite sides of the cell, ensuring faithful DNA segregation.

The mechanism of DNA segregation from type I partition systems is not fully understood [2]. Nevertheless, the DNA binding properties of ParA have been shown to play an essential role in segregation. In its ATP-bound form, ParA binds non-specifically to DNA [3,4,5], and this activity was shown to be essential for DNA partition in vivo [4,6]. Some ParA also display a specific DNA binding activity [7], provided by an additional amino-terminal (Nter) domain. This distinction defines two main subclasses, namely type Ia and type Ib [8]. Only type Ia ParAs bind DNA specifically to their promoter regions and function as transcriptional regulators of their own operons [9,10,11]. This autoregulation is enhanced by ParB proteins acting as corepressors of transcription [9,11] and is further enhanced by the centromere DNA [12,13]. The additional Nter domain is composed of ~100 aa and contains a winged helix-turn-helix (wHTH) motif [14]. The wHTH is involved in promoter-binding activity [15,16] when ParA is either in ADP-bound or ligand-free forms [5,17]. The role of the autoregulation of plasmid-encoded Par operons has been highlighted for a long time as both the absolute and relative levels of ParA and ParB are important for the partition process [18,19,20].

The partition system of *Escherichia coli* plasmid F, ParABS_F_ (historically called SopABC), belongs to type Ia. It encodes ParA_F_ and ParB_F_ proteins in an operon upstream of the *parS*_F_ centromere site (Figure 1). ParA_F_ binds to the promoter region of *parAB*_F_ (*P*parAB_F_) and ParB_F_ enhances this binding [7]. DNAse I footprint assays suggested that ParA_F_ binds to four copies of the hexamer motif 5′CTTTGC present in *P*parAB_F_. Three hexamer motifs are arrayed in direct orientation with 13-bp intervals, and one is present in an inverted orientation 28-bp further downstream (Figure 1). ParA_F_ binds to *P*parAB_F_ cooperatively and induces DNA bending [21]. It was proposed that ParA_F_ monomers bind to *P*parAB_F_ with a 4:1 stoichiometry, one on each hexamer motif [21]. However, how several ParA monomers bind in a coordinate fashion to these binding motifs is not known. Furthermore, results of our own investigations were inconsistent with this proposal and, in consequence, we have re-examined the mechanism underlying the autoregulation of type Ia plasmid partition operon.

In the present work, we further characterized ParA_F_ site-specific DNA binding properties on its promoter region. Using quantitative biochemical studies, we revealed a stoichiometry of three ParA_F_ dimers assembled on *P*parAB_F_ DNA. By modeling and docking a ParA_F_ dimer on the promoter region, we proposed that two hexamer motifs form a perfect inverted repeat (IR) binding motif separated by a large spacer. We also uncovered another similar but degenerated motif that overlaps the perfect IR. Molecular dynamics simulations suggested that the wHTH is a highly flexible region supporting the binding of ParA_F_ dimer to the overlapping motifs. Based on the *P*parAB_F_ DNA sequence and on closely related Par systems, we proposed that three ParA_F_ dimers bind in an overlapping manner to the promoter region. Finally, similar promoter organization was also found in other type Ia partition systems, suggesting that the proposed assembly for auto-regulating partition promoter may be conserved and widespread on low-copy-number plasmids.

## 2. Materials and Methods

### 2.1. Protein Purification and Analytical Gel Filtration

Native ParA_F_ and ParB_F_ proteins were purified from derivatives of strain DLT812 carrying pDAG127 (*P*ara_BAD_::*parA*_F_) [20] and pDAG170 (*P*ara_BAD_::*parB*_F_) [5]. Protein production was induced by adding arabinose (final concentration 0.1%) to 1 L cultures grown at 37 °C to OD_600_ ~ 0.6 in LB supplemented with thymine (10 µg·mL^−1^) and kanamycin (50 µg·mL^−1^). Four hours after induction, cells were harvested by centrifugation, resuspended in cold buffer A (50 mM NaCl, 1 mM EDTA, 20% sucrose), equilibrated at pH 6 (50 mM bis-Tris) for ParA_F_ or pH 8 (50 mM Tris) for ParB_F_. Cells were frozen at −80 °C and slowly thawed on ice. All subsequent steps were performed at 4 °C. Cell-free extracts were obtained by sonication followed by centrifugation for 20 min. at 27,000× *g* and filtration of the supernatant through a 0.2 µm hydrophilic filter (PALL). The filtrates were loaded onto 10 mL Heparin column on a FPLC (GE Healthcare) equilibrated in buffer A without sucrose. After washing with at least 100 mL of buffer A, the proteins were eluted over a 50–500 mM NaCl gradient. Fractions containing ParA_F_ or ParB_F_, identified by SDS-PAGE, were pooled. Ammonium sulphate was added to concentrations of 0.32 mg·mL^−1^ and 0.15 mg·mL^−1^ for ParA_F_ and ParB_F_, respectively, and the solutions were incubated for 30 min prior to centrifugation for 20 min at 20,000× *g*. The supernatants were loaded on a Superdex 200 column (GE Healthcare), equilibrated in buffer C (200 mM NaCl, 0.5 mM EDTA) and equilibrated at pH 6 for ParA_F_ (40 mM bis-Tris) or pH 8 for ParB_F_ (40 mM Tris). Fractions containing ParA_F_ or ParB_F_ were pooled, diluted two-fold in buffer C without NaCl, loaded onto a mono-Q or a mono-S column (GE Healthcare) and equilibrated in buffer D (buffer C containing 100 mM NaCl) for ParA_F_ or ParB_F_, respectively. After elution over a 100–600 mM NaCl gradient, each fraction containing ParA_F_ or ParB_F_ was mixed with cold glycerol (50 % final) and DTT (5 mM). All fractions were flash frozen in liquid nitrogen and stored at −80 °C.

Protein concentrations were determined from absorbance at 280 nm using a Nanodrop spectrophotometer. The ParA_F_ and ParB_F_ preparations were pure to >97% homogeneity as judged by SDS-PAGE stained by Instant Blue (Euromedex). An example of ParA_F_ purification was shown in Appendix A.

For analytical gel filtration, ParA_F_ diluted in buffer E (50 mM bis-Tris pH 6, 400 mM NaCl, 0.5 mM DTT) to 4 µM, in the presence or absence of ADP (1 mM), was injected onto a calibrated Superdex 75_10/300GL column (GE healthcare) and eluted at 0.5 mL·min^−1^ in the same buffer.

### 2.2. Surface Plasmon Resonance Assays

SPR experiments were performed using a Biacore 3000 (Biacore) or a Biacore X100 (GE Healthcare) apparatus. 200–530 RU (indicated in the Figure legends) of biotinylated DNA probes (Appendix A) were immobilized on a streptavidin-coated sensor chip in HBS-EP buffer (10 mM Hepes pH 7.4, 150 mM NaCl, 3 mM EDTA, 0.005% P20). Binding analyses were performed at 24 °C with multiple injections at different protein concentrations. Samples, diluted in BD buffer (20 mM Hepes-KOH pH 7.4, 100 mM KCl, 10 mM MgCl_2_, 1 mM DTT), were injected at 20 µL·min^−1^ for 180 s. Reference sensorgrams containing non-specific DNA (Biacore 3000) or no DNA (Biacore X100) were subtracted from sensorgrams containing the tested probes to yield binding responses. No significant difference was observed between the two methods of subtraction. Kinetic constants were calculated using BIAevaluation 4.0.1 software (Biacore, Uppsala, Sweden) taking into account survey’s recommendations. Values of *K_D_*, *k_a_* and *k_d_* were derived from local analysis fit with the 1:1 Langmuir binding model. *R_max_* is the theoretical maximum response of binding to the ligand at a specific surface density of analytes. The kinetics values were variable depending on experiments and apparatus, preventing their accurate determinations as indicated in the result section. Nevertheless, SPR assays allowed for accurately calculating the precise measurement of binding stoichiometry.

### 2.3. Electrophoretic Mobility Shift Assays

The standard binding mixture (10 µL) contained 15 nM Cyanine 3 (Cy3)-labeled DNA probes in 30 mM HEPES-KOH (pH 7.5), 150 mM KCl, 3 mM MgCl_2_, 1 mM DTT, 10% glycerol and, where indicated, 1 mM ADP. The mixtures, assembled on ice, at the indicated protein concentration, were incubated for 15 min at 37 °C and analyzed by electrophoresis at 50 V for 4 h or 200 V for 1 h at 4 °C on 6% polyacrylamide gels in TGE (25 mM Tris, 25 mM Glycine and 5 mM EDTA). Wet gels were scanned at 532 nm using a Typhoon trio imager (GE Healthcare, Chicago, IL, USA). Percentages of bound and unbound fractions were quantified using TINA software (Budapest, Hungary). Each experiment was performed at least three times. Data were subjected to non-linear regression analysis using GraphPad Prism 4.0 Software (San Diego, CA, USA).

### 2.4. Microscale Thermophoresis Assays

Microscale thermophoresis measurements were performed at 25 °C using Monolith NT™ premium capillaries and the Monolith NT.115 device with a green filter (NanoTemper, Munchen, Germany). Power of LED and IR-laser sources are indicated in figure legends. Binding reactions, assembled in low binding grade tubes (Eppendorf), consisted of MST buffer (10 mM Tris-HCl pH 8, 150 mM NaCl, 50 mM CaCl_2_, 0.05% Tween 20, 20 mg·mL^−1^ BSA), 25 nM Cy3-labeled, annealed oligonucleotide probes (Appendix A) and increasing concentrations of ParA_F_. Proteins and DNA were centrifuged prior to use. Fluorescence response was measured once total fluorescence reached equilibrium in all capillaries (SDS denaturation was undertaken following NanoTemper recommendation). Thermophoresis data were analyzed when fluorescence signals lay within the 10% acceptance range using MO Affinity Analysis software V2.1.2 (NanoTemper, Munchen, Germany). Data were plotted using GraphPad Prism 4.0 software (GraphPad Software, San Diego, CA, USA) and fitted with nonlinear equation.

### 2.5. Protein Melting Point Analysis

Protein thermal stability was measured in a label-free fluorescence assay using a Prometheus NT.48 (NanoTemper, Munchen, Germany). Briefly, the shift of intrinsic tryptophan fluorescence upon temperature-induced unfolding was monitored by detecting the emission fluorescence at 330 and 350 nm. Thermal unfolding was performed in nanoDSF grade high-sensitivity glass capillaries (NanoTemper, Munchen, Germany) at a heating rate of 1 °C per minute. Melting points (Tm) were calculated from the first derivative of the ratio of tryptophan emission intensities at 330 and 350 nm. A screen was performed over 18 different buffers. The melting temperature varies from 34.4 °C up to 49.5 °C (average 44.5 °C).

### 2.6. Modeling of ParA_F_ Structure and Docking on DNA

A three-dimensional model (3D) of ParA_F_ UniProt P62556 (SOPA_ECOLI) was built by homology modeling using Swiss-Model and Expasy with the X-ray structure of ParA_P1_ as a template (PDB 3ez2) [14]. We also modeled ParA_F_ using I-TASSER and Phyre2 [22,23]. Automatic docking of DNA on the 3D model of ParA_F_ was carried out using HADDOCK 2.2 [24]. Curved DNA models were obtained using 3D DART [25] by applying different angles between each nucleotide. The best docking was observed with a 3° angle given rising to an overall curvature of 120° over a 40-bp DNA fragment. The automated prediction of hinges in ParA_F_ structure was performed by HingeProt analysis [26]. Secondary structures were analyzes using hydrophobic cluster analysis (HCA [27]) and IUPred analyses [28].

### 2.7. Molecular Dynamics Simulation

From the 3D model of ParA_F_, molecular dynamics simulations were carried out using the AMBER 14 suite of programs and the molecular all-atom ff14SB force field [29]. Parameters for ADP were derived from [30]. To obtain a neutral charge of the simulated system, a number of Na^+^ counter-ions were included. Proteins together with the counter-ions were solvated with TIP3P water molecules, using the rectangular parallelepiped box with a minimum distance of 0.10 nm between the solute and the simulation box edge. Preparation of simulations consisted of initial energy minimization steps (steepest descent and conjugate gradient methods). Positions of the protein backbone and ADP atoms were first restrained using a harmonic potential during the minimization schedule. The force constant was then progressively diminished until a final unrestrained minimization step. The minimization process was followed by a slow heating under constant volume over a period of 100 ps. At the final temperature (310 K), the system was equilibrated under a constant volume condition over 10 ps and then subjected to constant pressure (1 bar) condition over 90 ps. The final production phase of simulations was then carried out for a total of 40 ns at constant temperature (310 K) and pressure (1 bar). Temperature and pressure parameters were set up using a Berendsen thermostat and barostat with a collision frequency of 2 ps^−1^ and pressure relaxation time of 2 ps. Long-range electrostatic forces were handled by using the particle-mesh Ewald method. The time step of the simulations was 2.0 fs and the SHAKE algorithm was used to constrain the lengths of all chemical bonds involving hydrogen atoms to their equilibrium values. To avoid artefacts, MD simulations were run twice, with different starting velocity distribution. The resulting trajectories were analyzed using the *cpptraj* module of the AMBER14 package. The RMSD was calculated for the protein backbone atoms using least squares fitting. Positional fluctuations (*Δ_ri_^2^*) of protein backbone atoms were calculated. A mass-weighted average value was then calculated for each residue. These parameters are related to the B-factors (depicting the atomic fluctuation) through the following relationship: $Bi=\frac{8\pi r^{2}}{3}\langle \;∆r_{i}^{2}\rangle$. The simulated B-factors were calculated using the coordinates of the 40 ns trajectory. All graphics were prepared using PyMOL (New York, NY, USA) [31].

## 3. Results

### 3.1. Three ParA_F_ Dimers Bind to the Para Promoter Region

To investigate how ParA_F_ interacts with its promoter, we performed a combination of three techniques measuring protein-DNA interactions. We first characterized this interaction using Surface Plasmon Resonance (SPR) measurements with increasing concentrations of ParA_F_ injected onto immobilized 136-bp *P*parAB_F_ DNA substrate. We obtained typical dose−response binding curves up to signal saturation above 500 nM (Figure 2A). Similar results were obtained in the presence of ADP (Appendix A), showing that the ADP-bound and apo forms of ParA_F_ display the same DNA binding activity, as previously observed [5]. No SPR signal was detected if *P*parAB_F_ DNA was replaced by non-specific DNA probes of the same length, indicating that ParA_F_ binding to the promoter region is highly specific in the absence (Figure 2B) or presence of ADP (Appendix A). SPR allowed direct and accurate calculation of the stoichiometry of ParA_F_ per DNA substrate as (i) the signal response is directly proportional to the molecular weight of the complexes at the surface of the sensor chip, where a given and known amount of DNA is immobilized, and (ii) SPR only measures the active protein present in the protein preparations. We estimated the stoichiometry of ParA_F_ per DNA substrate by measuring the ratio of the SPR signal response relative to the amount of *P*parAB_F_ DNA immobilized (Figure 2C, left panel). We found that six ParA_F_ monomers bind per *P*parAB_F_ DNA fragment when saturation is achieved. The same stoichiometry was also observed in the presence of ADP (Appendix A).

In the case of plasmid P1, apo-ParA_P1_ was shown to be mostly a dimer in solution [14,32]. To investigate the oligomeric state of ParA_F_, we performed analytic exclusion chromatography. At nearly physiological ParA_F_ concentration (4 µM), we observed only one peak, in the presence or absence of ADP, which corresponds to a molecular weight slightly above 80 kDa (Figure 2D). Thus, this indicates that ParA_F_ is mostly a dimer in the apo and ADP-bound forms. In the above SPR experiments, the stoichiometry is therefore three ParA_F_ dimers per *P*parAB_F_ promoter (Figure 2C). This result differs from a previous report using electrophoretic mobility-shift assay (EMSA) which concluded that four ParA_F_ monomers bind to the four 5′CTTTGC sequences of *P*parAB_F_ [21]. To discriminate between these contradictory results, we performed SPR analyses in other conditions such as in the presence of competitor DNA or non-hydrolysable ATP (ATPγS, which conferred a similar conformation of ParA_F_ as in the presence of ADP [6]), and we observed the same stoichiometry of three dimers per DNA fragments (Figure 2C, right panel). Importantly, this 3:1 stoichiometry was also observed with a 240-bp DNA fragment (Figure 2C, right panel) indicating that the size of the DNA probe does not limit the amount of ParA_F_ bound to the promoter region.

These experiments, reproduced on different Biacore apparatus with the same stoichiometry outcomes (Appendix A), displayed a large variability in the SPR binding isotherms (Appendix A), thus preventing to determine accurately the ParA_F_-*P*parAB_F_ dissociation constant (*K_D_*). By contrast, microscale thermophoresis (MST) assays and direct fluorescent measurements yielded a *K_D_* of 140 and 130 nM, respectively, for ParA_F_ binding to a 120-bp Cy3-labeled *P*parAB_F_ DNA probe (Figure 3A, upper panel; Appendix A). MST measurements performed in the presence of BSA, PEG or ADP resulted in similar *K_D_* ranging from 60 to 200 nM.

Lastly, we performed EMSA with the 120-bp Cy3-labeled *P*parAB_F_ fragment (Figure 3B). A unique band shift was obtained from 30 nM of ParA_F_. No band shift was observed when the DNA fragment did not harbor the motif #4, whatever the combination tested (Figure 3C). However, a band shift was observed when at least motifs #3 and #4 were present. The binding curve from the 120-bp *P*parAB_F_ fragment, fitted with a nonlinear regression, resulted in an apparent *K_D_* of 85 ± 20 nM (Figure 3D) with a sharp slope, suggesting high cooperativity as previously observed [21]. With longer incubation times, some ParA_F_ appeared stacked in the wells (judged by silver staining) at concentrations above 0.5 µM, indicating that some level of aggregation occurs at high concentrations, as also observed in the other techniques used (see figure legends). This effect is less pronounced in the presence of ADP (discussed below). Nevertheless, combining the results obtained from these different techniques, we estimated the *K_D_* of the interaction of *P*parAB_F_ promoter with three ParA_F_ dimers to be in the order of 100 nM and that motifs #3 and #4 are the primary ParA_F_ binding sites.

### 3.2. Modeling ParA_F_ Structure from ParA_P1_

Understanding how three ParA_F_ dimers interact with the promoter region would benefit from the knowledge of the 3D structure. Lacking an experimentally-determined structure of ParA_F_, we built a 3D structural model by comparative modeling using the 3D coordinates of the ADP-ParA_P1_ structure [14]. The two proteins share strong functional features and all deviant Walker-A ParA ATPases display important structural similarities [3,14,33]. We first characterized the extent of their secondary structure similitudes. Despite ParA_F_ and ParA_P1_ sharing low sequence identity and similarity (22% and 54%, respectively), the predicted ParA_F_ secondary structure is similar to the one of ParA_P1_ obtained by X-ray crystallography (Figure 4A). ParA_F_ contains two main parts discerned by hydrophobic cluster analysis (HCA) and IUPred analyses (Appendix A). The first part (aa 1–110) has the lowest structuration of the protein and encompasses the dimerization and the wHTH domains. The first Nter α-helix (aa 6–27) may be important for ParA_F_ dimerization, as shown for ParA_P1_ [14]. This is further supported by our failure to purify an active variant that lacks this part (Vergne, Castaing and Bouet, unpublished). In addition, HingeProt analysis predicted that two hinges separate the wHTH domain from the first α-helix from (position 30) and the second part (position 105) (Figure 4A and Appendix A). The second part (aa 111–388) contains the main conserved residues [8], and it comprises the deviant Walker-box and the ns-DNA binding domains (Figure 4A). Notably, the last α-helix (aa 367–388) is not conserved amongst ParA members.

These numerous similitudes between ParA_F_ and ParA_P1_ allowed us to build a ParA_F_ 3D model (Figure 4B) by homology modeling using three different approaches (SwissModel, Phyre2 and I-tasser). The RMSD value that compares the template and the best model backbone was ~1 Å, indicating a good modeling accuracy. Besides this structural analysis, we attempted to construct several variants in the Nter domain and the wHTH motif specifically impaired for *P*parAB_F_ binding. However, all deletion variants tested were poorly expressed, suggesting that these proteins were not stable. Moreover, two variants in the wHTH (A46P and R57A-K61A) were purified, but melting point analysis using nanoDSF (advanced Differential Scanning Fluorimetry) suggests that they were significantly unstructured, thus preventing further studies.

Overall, these results indicated that ParA_F_ and ParA_P1_ display predicted structural similarities and that the first 110 amino acids of ParA_F_ including the wHTH are important for correct folding.

### 3.3. Overlapping Binding of Three ParA_F_ Dimers in the Promoter Region

Next, by careful examination of the *P*parAB_F_ sequence, we found that the 5′CTTTGC motifs #3 and #4 (Figure 1) may form an inverted repeat (IR) motif with a large 28-bp spacer that would accommodate the binding of a ParA_F_ dimer. We investigated this hypothesis by docking a ParA_F_ dimer onto this putative IR motif (Figure 5A). The distance between the centers of the two binding motifs is 34-bp; therefore, they are both located nearly on the same face of the DNA molecule. Each motif could interact with each wHTH, separated by ~10 nm, of the ParA_F_ dimer. However, such binding could only be possible if the DNA is curved by 3° between each base pair, as deduced from docking trials (Figure 5A). Interestingly, the *P*parAB_F_ region is AT rich (73% over 100-bp), presents A-stretches (a property which is known to be prone to induce DNA curvature [34]) and is bent upon ParA_F_ binding [21]. Moreover, the modeling of ParA_P1_ dimer bound to a 40-bp DNA fragment has also suggested that DNA bending is required to allow the two wHTH to contact the DNA major groove [14]. In the following, we will refer to this motif as IR3-4 (Figure 5C).

We next extended our analyses to the binding motif #2. We found the similar but slightly degenerated sequence, 5′tcTTGC, 27-bp away in an inverted orientation (Figure 5C), which we named #2′ and which could form a putative IR2-2′. For binding motif #1, we could not find a complementary inverted repeat ~28-bp away since this position exactly overlaps the −35 promoter box. Previous DNase I footprint experiments performed on *P*parAB_F_ DNA [7,21] have shown that ParA_F_ protection extends over 87-bp (schematically represented in Figure 5C), starting and ending in the close vicinity of binding motifs #1 and #4, respectively. Importantly, a protection was observed over the imperfect binding motif #2′ [21]. Combining these results with the stoichiometry experiments, we propose that ParA_F_ dimers could bind to IR3-4, IR2-2′ and motif #1, the latter being bound only through one wHTH of a dimer, leading to three dimers bound to *P*parAB_F_ as observed by SPR (Figure 2C). This binding organization implies that ParA_F_ dimers are overlapping each other to contact interspersed binding motifs present two by two in inverse orientations on the same face of the DNA molecule.

We then examined promoter regions of three partition systems closely related to the one of plasmid F, the *K. pneumoniae* virulence plasmid pLVPK [37], the *E. coli* linear plasmid-prophage N15 [38] and the *K. oxytoca* plasmid-prophage phiKO2 [39]. In all three cases, while the sequences in between the nearly identical binding motifs vary, they display the same spacing between each other (Figure 5C). Notably, for N15, the binding motif #2′, present in inverted orientation with a 28-bp spacing from the motif #2, has only one variation from the consensus (5′GtAAAG). This strengthens the hypothesis that these two hexamer sequences form an IR to which a ParA_F_ dimer could bind. We performed BLASTn searches with the Par promoter regions of F, LVPK, N15 and KO2, and found 12 other closely related, but not identical, partition promoters (Appendix A). We carried out a quantitative Logo analysis using these 16 sequences (Figure 5D). The conserved pattern still emerged clearly both in terms of sequence and spacing between each binding motif. This finding further argues for a binding model involving overlapping IR motifs in these partition promoters.

### 3.4. Is the Promoter Organization with Overlapping IR Motifs a Conserved Feature amongst Type Ia Plasmid Partition Operons?

To evaluate whether a promoter organization with overlapping IR binding motifs is a general or restricted feature, we extended the comparative analysis to more phylogenetically distant partition systems. The promoter regions of the Par operons of plasmids P1 (*P*parAB_P1_) and P7 (*P*parAB_P7_) display a different organization from the one of plasmid F. They both harbor a large degenerated palindromic sequence but with different positioning relative to their respective −35 and −10 boxes (schematically represented by black arrows in Figure 5E) and also, in *P*parAB_P7_, a set of four 9-bp imperfect repeats (open arrows) [10,38]. In *P*parAB_P1_, we found a 6-bp motif, 5′TTATGC, at both extremities of the large palindromic sequence, in an inverse orientation with a 28-bp spacing (IR3-3′; Figure 5E), which could accommodate the binding of a ParA_P1_ dimer as proposed above for ParA_F_. Strikingly, we found two other sets of similar sequences in inverse orientation with 26- and 28-bp spacing. As for F, only one IR motif (IR3-3′) is perfectly conserved, the two others (IR1-1′ and IR2-2′) being imperfect. The binding motif #2, 5′acAaGC, is the less conserved but overlaps the −35 promoter box. The logo analysis of the six proposed binding motifs in *P*parAB_P1_ indicates that four out of six nucleotides are highly conserved as for the motifs in *P*parAB_F_ (Figure 5F). Overall, the layout of these three putative IR motifs is highly reminiscent to the one of *P*parAB_F_, with only IR3-3′ being shifted downstream 8-bp, respectively, in correlation with the −35 and −10 boxes being shifted upstream, compared to *P*parAB_F_. We found that the position of these proposed IR motifs matches the protection zones of previous DNAse I footprinting experiments [32,36,40], which begins at the motif #1 and ends a few bp after motif #3′ (Figure 5E, grey lines). Moreover, DNAse I footprinting performed with a version of *P*parAB_P1_ lacking the DNA sequence upstream of the −35 promoter box still revealed a protection at the position corresponding to the missing binding motif #1, 5′gcATGC [32]. Thus, only one binding motif of the IR (in this case #1′) is required to allow ParA_P1_ dimers to efficiently protect the upstream DNA sequence (#1) by a non-specific DNA binding interaction with the second wHTH.

In the P7 Par promoter, *P*parAB_P7_, a perfect IR motif composed of the sequences 5′ACGTGC separated by 29-bp also encompassed the imperfect palindromic sequences previously proposed [10]. Two other sets of similar motifs matching a consensus 5′ACGTgc present two by two in inverted orientation with 26- and 27-bp spacers were present in *P*parAB_P7_ (Figure 5E). However, in contrast to *P*parAB of F and P1, only the proposed motifs IR1-1′ and IR2-2′ overlapped; the putative IR3-3′being shifted 21-bp downstream. This latter is also less conserved which leads to a slightly lower signature for a binding motif in the logo analysis (Figure 5F). The overlapping motifs IR1-1′ and IR2-2′ correspond perfectly to the ParA_P7_ protected zones in DNAse I footprinting (Figure 5E, grey bars), further arguing for being the binding motifs. In the case of IR3-3′, a DNA protection is seen close to the motif #3 but not at the proposed motif #3′, which raises the question of whether this latter IR motif is involved in autoregulation of *P*parAB_P7_.

These analyses support the possibility that, as for *P*parAB_F_, the partition promoters of P1 and P7 harbor IR motifs with 26- to 29-bp internal spacing that may be involved in the binding of their cognate ParA dimers. Interestingly, with the only exception of IR3-3′ from P7, these motifs are similarly positioned relatively to each other in these three different promoter regions (Figure 5C–E). This finding suggests that some type Ia promoters of phylogenetically distant plasmid-encoded partition operons may share this overlapping binding organization.

### 3.5. The Winged-HTH Domain Is Highly Flexible

The analysis of ParA_F_ and ParA_P1_ amino acid sequences and structures has revealed a conservation of secondary and tertiary structures. An important feature of the type Ia plasmid ParA proteins is their additional Nter domain comprising the wHTH. Our observation suggests that the *P*parAB_F_ region harbors inverted repeats where three ParA_F_ dimers could bind if the DNA is curved (Figure 5A). Another possibility would be that the protein is flexible and undergoes conformational changes, enlarging its amplitude. We investigated how the wHTH region participates in the specific DNA binding by analyzing the conformational plasticity of ParA_F_ protein. To this end, we performed Molecular Dynamics (MD) simulations in explicit water environments. The stability of ParA_F_ dimers during the course of the MD simulation was evaluated by backbone atoms RMSD measurement after least square fit from the starting structure (Appendix A). The profile obtained indicates that the backbone RMSD gradually increased by 7 Å within the first 7 ns of the simulation and then remained stable until the end of the simulation. Such RMSD variation indicates thus that the protein structure had undergone significant conformational changes (Appendix A). The analysis of the B-factors during the course of MD simulations highlights particularly flexible regions of the protein (Figure 4C). They are predominantly located in the wHTH region involving α2–α3 helices and β2–β3 antiparallel β strands (Appendix A). In particular, the glutamic acid and the arginine residues at positions 60 and 75, respectively, exhibit the highest simulated B-factor values and correspond to the end of the HTH (E60) and to the inflexion point of the winged (R75). It is noteworthy that ParA_P1_ also presents high simulated B-factors within the wHTH domain (centered around residue 75; Appendix A).

The backbone superposition of the starting ParA_F_ structure, with the one obtained after 40 ns of MD simulation, shows that the arginine residue R75 can move ~30 Å away from its initial position (Figure 5B), while the core domain remains stable (Appendix A). Notably, the conformational flexibility enlarges the wing span and increases the width of ParA_F_ dimer from 10 nm to 11 nm. This increase of protein amplitude may be needed for contacting the inverted repeats motifs. This result is also in agreement with the prediction of two Hinges at positions 30 and 105 (Figure 4A). Overall, the MD simulations show that the wHTH region is highly flexible and may undergo important variation in amplitude movement, increasing the wingspan of ParA_F_. We propose that, in addition to DNA bending, conformational changes may adjust ParA_F_ dimers binding to the IR motifs to support local bending and topological constraints.

### 3.6. The IR3-4 Motif Is Sufficient to Nucleate the Binding of Three ParA_F_ Dimers

The minimal size of the DNA fragment required for three ParA_F_ dimers to bind to *P*parAB_F_ was investigated by SPR assays. We designed DNA duplexes containing at least the IR3-4 motif (Appendix A). No response signal was obtained in SPR assays using a 58-bp DNA fragment, indicating that a minimal size is required for ParA_F_ binding. By contrast, an SPR response was observed with an 85-bp probe. Despite that the concentration of ParA_F_ needed to reach saturation was significantly higher (5 µM) than for the larger DNA fragments, a stoichiometry of 2.7 was calculated (Figure 2C, right panel). This indicates that the IR3-4 motif is sufficient to nucleate a complex with three ParA_F_ dimers on a DNA fragment as small as 85-bp.

Strikingly, we found that the position of the IR motif on small DNA fragments is crucial for ParA_F_ binding. Indeed, using EMSA, we observed that if the IR3-4 motif is positioned in the middle of the 120-bp DNA fragments (probe IR2-2*), no band shift was observed (Figure 3C). The same result was obtained if the motif #4 was also present (probe IR2-2*4). This result indicates that there is not enough space for a complex of three ParA_F_ dimers to bind starting from an IR motif positioned in the middle of the DNA fragment. Thus, these data confirm that the IR3-4 motif is sufficient to nucleate the binding of three ParA_F_ dimers to *P*parAB_F_ DNA, providing that the length of the DNA available on the left is large enough allowing stable interaction.

### 3.7. ADP Stabilizes ParA_F_ but Does Not Increase Its Affinity to PparAB_F_

ParA_P1_ was shown to interact preferentially with its promoter region in the presence of ADP [40,41]. For ParA_F_, no significant difference in binding affinity to *P*parAB_F_ was detected in the presence or absence of ADP in SPR and EMSA experiments (compare Figure 2A,B and Appendix A; Appendix A). However, in MST experiments, we observed that ADP stabilized the thermophoresis signals and limited ParA_F_ aggregation. To investigate this further, we performed protein melting point analyses using nanoDSF in different conditions (Table 1). In the presence of ADP, the ParA_F_ melting temperature (Tm) increased by 1.5 °C. By contrast, in the presence of *P*parAB_F_ DNA, ParA_F_ is stabilized only moderately with an increased Tm by 0.5 °C, and this effect is not cumulative with the ADP one (Table 1). Thus, ADP stabilizes ParA_F_ dimers compared to the ligand-free form. This may prevent the aggregation that occurs over time, but ADP does not increase ParA_F_ affinity to *P*parAB_F_.

### 3.8. ParB_F_ Increases ParA_F_ Binding Affinity to the Promoter Region

Early experiments using DNAse I footprinting assays have suggested that ParB_F_ increases the binding affinity of ParA_F_ to *P*parAB_F_ [7]. To quantify this stimulatory effect, we performed EMSA in the presence of both ParA_F_ and ParB_F_. ParB_F_ has no specific affinity for *P*parAB_F_ DNA [7] and presents a *K_D_* of ~1 µM to non-specific DNA probes (Appendix A) in agreement with previous measurements [42]. Maximal ParB_F_ stimulation of ParA_F_ binding to *P*parAB_F_ was obtained when the same stoichiometry of both proteins was used (Figure 3E). It resulted in an apparent *K_D_* of about 10 nM compared to 85 nM with ParA_F_ alone, corresponding to an eight-fold ParB_F_-dependent stimulation. Lastly, we also tried the addition of a duplex DNA containing *parS*_F_ in the presence of ParB_F_ and found no further stimulation of ParA_F_ binding.

## 4. Discussion

This study provides a new picture of how the partition operon, *parAB_F_*, is transcriptionally auto-regulated. It establishes that ParA_F_ is a dimer in solution and binds primarily to an IR motif. This first interaction allows two other dimers to bind, in a cooperative and oriented manner, to other overlapping motifs surrounding the −35 and −10 transcriptional boxes. The resulting complex composed of three dimers covers an 85-bp region. We propose that the cooperativity is mediated by protein−protein interaction through the highly flexible winged-HTH domain and that this binding organization may be conserved for some plasmid partition promoters of the type Ia subfamily whose finely-tuned expression is auto-regulated by ParA. Our findings are summarized in Figure 6.

Schematic representation of three ParA dimers bound over the *P*parAB promoter region. DNA is represented by a grey bar with the ParA binding motifs displayed by rectangular boxes and arrows colored according to Figure 5C. Each ParA monomer is represented by a semi-oval (Walker-box domain) link to an oval (Nter domain) with the same color-code as their respective binding motifs. ParA are dimers in the apo and ADP-bound forms that are stabilized by the first Nter α-helices (represented by curved lines ending by a zigzag), which maintained the two monomers together. The wHTH motifs, symbolized by the ovals with loops (drawn with darker lines) emanating from them, contact specifically the DNA binding motifs. A ParA dimer (green) binds, through the two symmetrically oriented wHTHs, to an inverted repeat motif composed of two perfect binding sites spaced by 28-bp (for F, IR3-4). This initial binding enables the cooperative loading of a second ParA dimer (orange) on imperfect IR motif (IR2-2′) composed of a slightly degenerated motif #2′ in reverse orientation and spaced by 27-bp relatively to the motif #2, and of a third dimer (blue) on an imperfect IR motif (for F, composed of only 1 perfect binding site (#1)). We propose that these two secondary bindings are mediated by dimer−dimer contacts through the wings of two intertwined dimers, resulting in a stable complex composed of three dimers bound to the promoter region. The flexible wHTH allows each monomer to contact one hexamer motif by adapting to the local DNA constraint and to interact with an adjacent dimer bound to the overlapping binding motifs through the wing. The overall complex assembled on the promoter region covers ~85-bp. For simplicity, the schematic is drawn with a straight representation and not with bent DNA. ParB may increase ParA binding affinity by enabling an extended conformation proficient for the assembly of a stable complex without interacting directly on the complex.

ParA proteins of type I partition systems are dimers in the presence of ATP [3,32]. In contrast to chromosomally-encoded ParAs (belonging to type Ib [8], which are converted to monomers upon ATP hydrolysis [3]), type Ia ParAs remain mainly dimers, as exemplified by ParA_P1_ and ParA_P7_ [14,32] and shown for ParA_F_ (Figure 2D). The first α-helix of the Nter region of ParA_P1-P7_ stabilizes the dimer form [14]. This α-helix was conserved on ParA_F_ (Appendix A) and 3D-modeling strongly suggests that it may exhibit the same role in dimer stabilization as for ParA_P1-P7_ (Figure 4B). The Nter domain is not present on type Ib ParAs [8], therefore explaining this difference in dimer to monomer conversion upon ATP hydrolysis between plasmid and chromosomally encoded ParAs. Moreover, the expression of partition operons is auto-regulated by ParAs only for type Ia plasmid members through the wHTH DNA binding in the Nter domain [10,12,16].

ParA_F_ specifically binds the promoter region in the apo and ADP-bound forms [5] as for ParA_P1_ [32,41]. Four ParA_F_ monomers were initially proposed to bind to the four hexamer repeats in the promoter region of *parAB*_F_ [21]. However, gel filtration experiments indicated that ParA_F_ is a dimer in solution, both in the presence or absence of ADP (Figure 2D), which rather argue, along with the quantitative SPR analyses, that three ParA_F_ dimers bind to the promoter region (Figure 2C). We thus hypothesized that ParA_F_ dimers may contact binding sites that would be present in inverted orientations and uncovered that the hexamer motifs #3 and #4 define a perfect inverted repeat motif with a 28-bp spacer. The discrepancy in EMSA between our results (Figure 3C) and a previous study may arise from the mutations in the binding motifs since we modified all 6-bp rather than only 3-bp [21]. This could explain why, in contrast to [21], we observed that mutating either motif #3 or #4 prevents ParA_F_ from binding to *P*parAB_F_. We also identified a slightly degenerated motif (#2′) in reverse orientation and spaced by 27-bp relative to the motif #2. These proposed binding sites are all protected by ParA_F_ in DNAse I footprinting assays (summarized in Figure 5C), including the motif #2′ [21]. In addition, mutations in motif #2 abolished the protection of the motif #2′ [21]. Therefore, these previous results together with our data strongly argue for ParA_F_ binding as a dimer contacting two DNA sites, separated by about three helical turn (~33-bp between the centers of each repeat). We propose that binding motifs #3 and #4 constitutes a perfect IR motif (IR3-4). Accordingly, motifs #2 and #2′ could form an imperfect IR motif (IR2-2′). These IR motifs are highly conserved in terms of sequences and spacing amongst closely-related promoters (Figure 5C and Appendix A). For motif #1, the position expected for a cognate inverted binding motif with ~28-bp spacing would be exactly at the position of the −35 box, which prevents a match with the consensus binding motif. Interestingly, the removal of the binding site #1 changed the footprint at the expected position for a cognate binding site [21], further arguing for ParA_F_ binding as a dimer even from a unique binding motif, provided that other IR motifs are present (see below).

Previous studies have investigated the promoter region of the Par operons of the plasmids P1 and P7 and reported that they are very different in sequence and organization [10]. Nevertheless, the 3D-structure of ParA from these two plasmids is highly similar and allowed a 3D-modeling of ParA_F_ with high accuracy (Figure 4B), thus suggesting that ParA binding as dimers to IR motifs in the promoter regions would also apply for P1 and P7. In both cases, as for F, we found two perfect 6-bp inverted motifs spaced by 28- or 29-bp and IR3-3′and IR1-1′ for P1 and P7, respectively (Figure 5E). Moreover, two other putative IR motifs with imperfect binding sites spaced by ~26- to 28-bp were present. These three promoter regions have different core sequences but display overlapping IR binding motifs with strikingly similar positioning relative to each other (Figure 5E). Therefore, we propose that the binding of a ParA dimer to the perfect IR motif serves as nucleating the cooperative binding of two other ParA dimers to the degenerated motifs, thus stabilizing the binding of three dimers to *P*parAB (see model Figure 6). This proposition also supports the unexplained observation that a truncated version of *P*parAB_P1_, lacking the DNA sequence upstream of the −35 promoter box, is still protected from DNAse I cleavage at the position corresponding to the missing binding motif #1 in a ParA_P1_-dependent manner [32]. For *P*parAB_F_, provided that at least one IR repeat motif is present, a ParA dimer could nucleate the binding of two other ParAs efficiently to cover the promoter region Figure 2C and Figure 3C). Altogether, these data suggest that the promoter structure of these type Ia plasmid partition operons is conserved from a putative common ancestry and also highlights the importance of the auto-regulation of the *par* genes’ expression.

ParA_F_ covers a large DNA region (~85-bp) encompassing the −35 and −10 core promoter sequences as observed in DNAse I footprinting assays [7,21]. This is in agreement with our findings that the minimal DNA fragment size for the binding of three ParA_F_ dimers is ~85-bp (Figure 2C, right panel). The binding affinity of ParA_F_ on the promoter region was measured to be in the order of ~100 nM (Figure 3). The binding curve displays a sharp slope (Figure 3D), suggesting an important binding cooperativity. Furthermore, mutations in any of the four conserved binding motifs do not change the overall protected zone [21], and the IR3-4 motif is sufficient to nucleate ParA_F_ binding to *P*parAB_F_, provided that DNA is longer than 85-bp (Figure 3C) and further arguing for the high binding cooperativity. Our data suggest that dimer−dimer interaction may favor the cooperative binding of three ParA_F_ dimers to *P*parAB_F_. Interestingly, wHTH motifs have been reported to be involved in protein−protein interaction [43], as in the case of the *B. subtilis* protein RacA [44], whose wHTH motif is the closest structure to that of ParA_F_. Moreover, a model prediction from the fitting of the ParA_P1_ crystal dimer (ADP form) into the nucleofilament of its close homolog ParA2*_Vcho_* from the chromosome 2 of *Vibrio cholera* also suggested that the wHTH makes contact with the N-terminal of other subunits [45]. In light of our results and the specific organization of the promoter region with overlapping binding motifs, it is therefore tempting to propose that a dimer−dimer interaction mediated by the wHTH motifs may allow ParA_F_ dimers to bind cooperatively on *P*parAB_F_ promoter (see model Figure 6). The wHTH is composed of a classical tri-helical bundle followed by two β-strand hairpins and one α-helix, forming an approximately triangular outline. The wing interacts with the minor groove of DNA through charged residues in the hairpin [43]. In the ParA_F_ dimer, the two wHTHs are present on both edges of the structure (Figure 4B). This symmetrical conformation allows binding to IR motifs. The two binding motifs of the proposed IR are separated by 33- or 34-bp (Figure 5C), corresponding to about three helical turns of 10.5-bp, and therefore they are present nearly on the same face of the DNA molecule. The 3D-modeling of a ParA_F_ dimer on a DNA fragment indicates that the distance between the two wHTHs motifs is shorter than the distance between the inverted binding motifs. Therefore, the specific interaction can occur only if the DNA is bent (Figure 5A), as proposed for ParA_P1_ [14] and observed on *P*parAB_F_ DNA upon ParA_F_ binding [21]. The promoter sequence contains numerous A-tracks known to promote important local bending of the DNA, thus suggesting that the DNA curvature is probably not homogeneous over the entire region. Our results based on molecular dynamics simulations suggest that the wHTH, insulated by two hinge motifs (Figure 4A), is the most flexible region of ParA_F_ (Appendix A). It could move up to 3 nm away from its initial positions (Figure 5B), expanding the overall span of the dimer by 10 angstroms. Therefore, this capability to change the amplitude between the two wHTH motifs and to adapt their positioning in relation to the local DNA constraint might be important for contacting the IR binding motifs and for interacting with a wHTH motif of another dimer (see the model in Figure 6).

ParB_F_ participates in the autoregulation of *parAB* expression [11] as also observed in the P1 Par system [9] and stimulates ParA_F_ binding to its promoter region [7]. We quantified this effect in vitro and observed a ~8-fold increase in binding affinity with a *K_D_* of ~10 nM, confirming that ParB_F_ stimulates ParA_F_ binding to the promoter region in the absence of ATP hydrolysis (Figure 3E). The action of ParB_F_ on ParA_F_ binding does not occur through a stable ParA−ParB interaction, since no super-shift of *P*parAB_F_-ParA_F_ complexes in EMSA or larger footprinting were observed upon addition of ParB_F_ [7]. In the case of plasmid P1, a specific enhancement of repression by ParB_P1_ was also shown to be due to a protein−protein interaction with ParA_P1_ rather than through a direct recognition of the operator by ParB_P1_ [15], in agreement with the observation that ParB_P1_ does not bind to the promoter region and does not cause any qualitative modification of the footprint of ParA_P1_ bound to it [46]. In the presence of the centromere site, *parS*_F_, the level of repression measured in vivo increased [12]. This also occurs when *parS*_F_ is present in *trans* on a different DNA molecule, thus indicating that the assembly of ParB_F_ on *parS*_F_ and surrounding sequences, including *P*parAB_F_, does not account for this additional repression level. One possible explanation for the stimulatory effect of ParB_F_ on ParA_F_ binding to its promoter region could be that a conformational change in ParA_F_ is induced by ParB_F_. In the case of plasmid P1, ParA_P1_ super-repressor variants are no more stimulated by ParB_P1_ to further increase repression [36]. It was suggested that these mutations, in the Walker A motif, might block ParA_P1_ in the conformation proficient for promoter binding [47]. We propose that a conformational change upon ParB_F_ interaction may extend the overall span of ParA_F_ and/or the relative orientation of the flexible wHTH motifs favoring the promoter-specific DNA binding affinity. A similar ParB-induced conformational change acting on ParA-ATP form, releasing ParA binding to non-specific DNA, has been previously proposed [47,48]. Further experiments are needed to determine if this ParB-induced conformational change is transient or requires the release of ATP before being converted back to the non-specific DNA binding conformation.

## 5. Conclusions

In conclusion, our results shed some light on key structural features of ParA_F_ dimers, leading us to propose a new organization with overlapping binding motifs in the promoter region. This organization is conserved on closely but also distantly related partition systems, suggesting that it may be important for the fine-tuning of the partition operon expression. The nature of the interactions between ParA_F_ dimers and their orientation relative to each other await future biochemical and structural studies.

## Figures and Tables

**Figure 1 genes-12-01345-f001:**
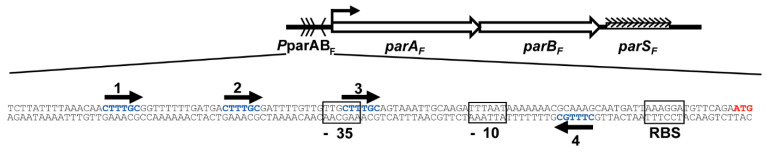
The *parABS*_F_ (*sopABC*) partition locus of the plasmid F. Top: Schematic representation of the *parABS*_F_ locus. The *parA*_F_ and *parB*_F_ genes, the promoter region (*P*parAB_F_) and the *cis*-acting *parS*_F_ site are drawn at scale over 3-kbp. The arrow represents the transcriptional start point. Arrowheads in *P*parAB_F_ display the positions and orientations of the hexamer 5′CTTTGC motifs. The arrowheads in the *parS*_F_ centromere site represent the twelve 43-bp tandem repeats containing the ParB_F_ binding sites. Bottom: Nucleotide sequence of the *P*parAB_F_ promoter region. The *parA*_F_ start codon and the hexamer motifs are labeled in red and blue, respectively. The −10 and −35 promoter motifs and the ribosome binding site (RBS) are indicated by boxes.

**Figure 2 genes-12-01345-f002:**
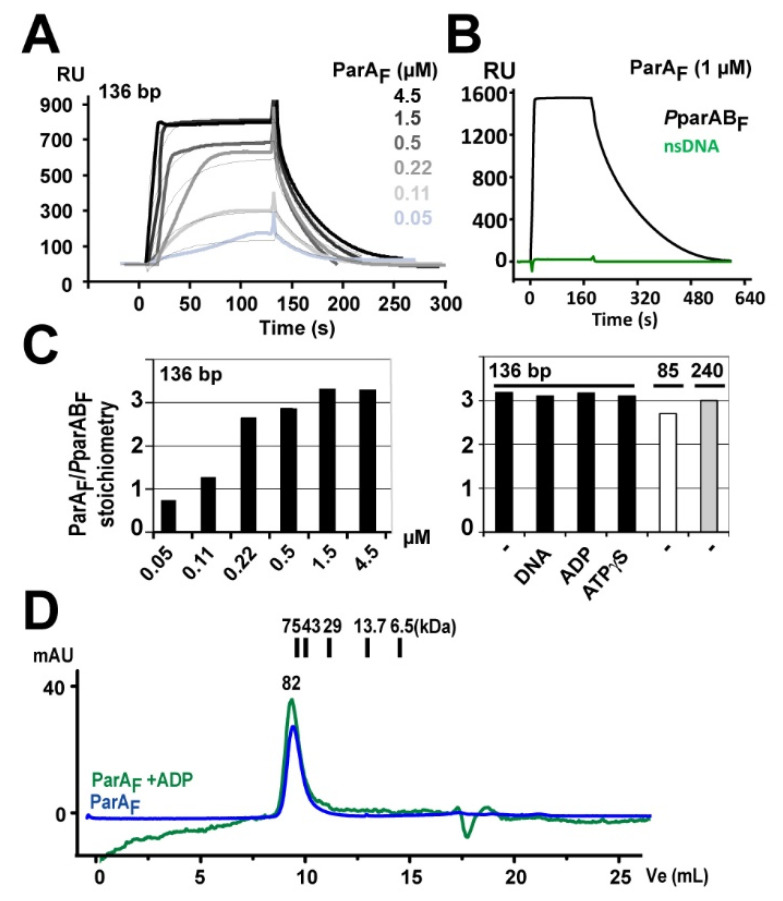
Three ParA_F_ dimers bind to the *P*parAB_F_ promoter. (**A**) Surface Plasmon Resonance analysis of ParA_F_ binding to *P*parAB_F_. 300 RU of a 136-bp biotinylated DNA fragment were immobilized on a streptavidin chip. Increasing concentrations of ParA_F_ ranging from 0.05 to 4.5 µM in BD buffer without adenine nucleotide were injected at 20 µL·min^−1^ at time 0. Dose−response sensorgrams and corresponding fits according to 1:1 Langmuir interaction model are represented by bold and light curves, respectively. (**B**) ParA_F_ binds specifically to the promoter region in the absence of ADP. ParA_F_ DNA binding activity was measured by SPR using 136-bp DNA fragments containing *P*parAB_F_ (530 RU; black line) or nsDNA (597 RU; green line) immobilized on two different channels of the same sensor chip. ParA_F_ (1 µM) was injected at 10 µL·min^−1^ at time 0 for 180 s before dissociation. (**C**) Stoichiometry of ParA_F_ dimers binding. ParA_F_ dimers bound per *P*parAB_F_ DNA molecule was calculated according to the formula S = *Rmax*/[(MW_A_/MW_L_) ∗ R_L_], where *Rmax* is the maximum response, MW_A_ is the molecular weight of ParA_F_ and MW_L_ and R_L_ are the molecular weight and the amount of ligand immobilized, respectively. ADP and non-hydrolysable ATP analog (ATPγS) were used at 1 mM and, where indicated, sonicated salmon sperm DNA is added at 1 mg·mL^−1^. Biotinylated DNA fragments of 85-, 136- and 240-bp, immobilized at 200, 300 and 530 RU, respectively, were infused with 5, 1.5 and 1.85 µM of ParA_F_, respectively. (**D**) ParA_F_ is a dimer in solution in the apo- and ADP-form. Analytical gel filtration chromatography of ParA_F_ (4 µM) pre-incubated with (green line) or without (blue line) 1 mM ADP using a Superdex 75 column was monitored at 280 nm. A drift of the baseline is observed throughout the elution in the presence of ADP, probably to a slight difference in the buffers used during the experiment. The elution peak (9.5 mL) corresponds to 82 kDa. Conalbumin (75 kDa), ovalbumin (43 kDa), anhydrate carbonic (29 kDa), ribonuclease A (13.7 kDa) and aprotinin (6.5 kDa) were used as molecular weight standards and eluted at 9.66, 10.55, 11.74, 13.35 and 15 mL, respectively.

**Figure 3 genes-12-01345-f003:**
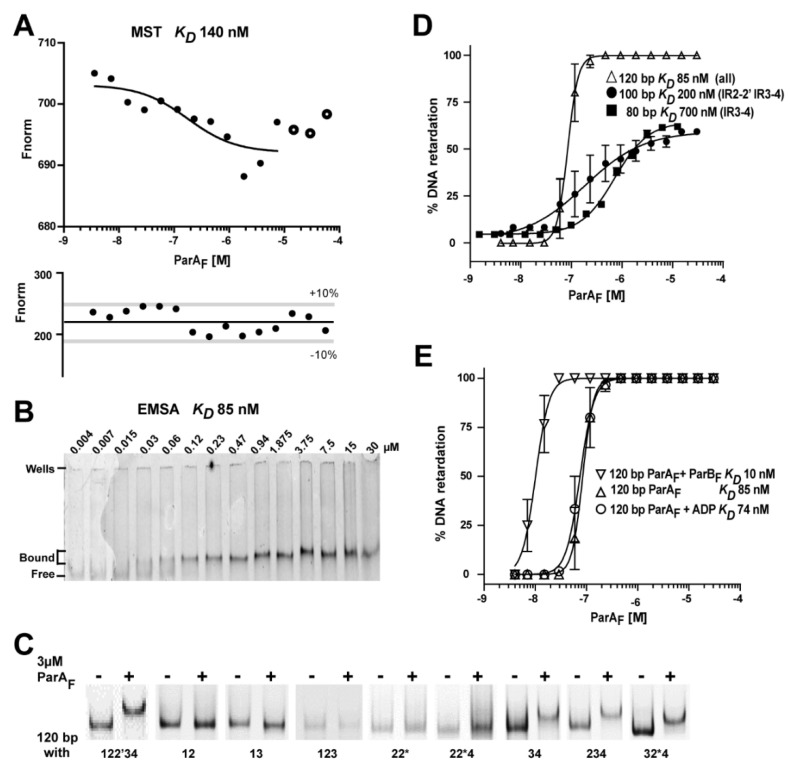
Measurements of ParA_F_ binding affinity to the *P*parAB_F_ promoter. (**A**) MST measurement of ParA_F_ binding to a 120-bp Cy3-labeled *P*parAB_F_ DNA fragment. DNA (25 nM) is titrated over ParA_F_ concentrations ranging from 3.6 nM to 60 µM. The apparatus power sources were set at 40% LED and 80% IR-laser. At high protein concentrations (above 15 µM), the decrease in the fluorescence signal is due to ParA_F_ aggregation (larger dots) and is not taken into account for curve fitting. *K_D_* was determined to be 140 ± 23 nM with a stable fluorescence signal in the 10% acceptance range (lower panel). (**B**) ParA_F_ binding to *P*parAB_F_ in EMSA. Cy3-labeled 120-bp DNA fragments (15 nM) carrying *P*parAB_F_ (with motifs 122′34) were incubated with increasing concentrations of ParA_F_ ranging from 4 nM to 30 µM. They were analyzed by electrophoresis in a 6% TGE polyacrylamide gel. The positions of the bound and free DNA fragments, and of the wells, are indicated on the left. (**C**) ParA_F_ binding to *P*parAB_F_ DNA fragments harboring mutations in the hexamer motifs. EMSA were performed as in B with 3 µM of ParA_F_ and various 120-bp probes carrying the indicated binding motifs (see Appendix A). Note that 2* refers to the 2′ motif replaced by a perfect consensus site. (**D**) ParA_F_ binding to *P*parAB_F_ probes of different lengths. Quantification of three independent EMSA experiments (except for the 80-bp DNA fragments assayed only twice), performed as in (**B**), with Cy3-labeled *P*parAB_F_ probes (15 nM) of various sizes containing IR3-4 and different combinations of the other binding motifs (see Appendix A). ParA_F_ protein concentrations range from 4 nM to 30 µM for the 120- and 100-bp probes and 1.5 nM to 12 µM for the 80-bp probes. Apparent *K_D_* were calculated after fitting with non-linear regression. (**E**) ParA_F_ binding to *P*parAB_F_ in the presence of ADP or ParB_F._, as in C, with Cy3-labeled 120-bp probes incubated with ParA_F_ concentrations ranging from 4 nM to 30 µM in the presence of ADP (1 mM) or ParB_F_ (equal molar concentration as ParA_F_). The apparent *K_D_* measured in the presence of ParB_F_ is an estimation since it is close to DNA probe concentration (15 nM).

**Figure 4 genes-12-01345-f004:**
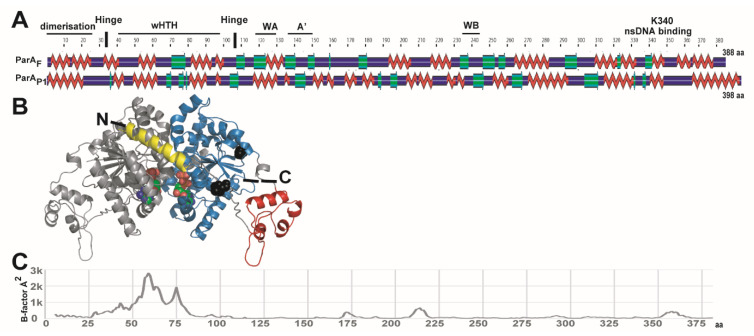
Structural similarity of ParA_F_ with ParA_P1_. (**A**) Alignment of secondary structures predicted for ParA_F_ (388 aa) and derived from X-ray crystallography for ParA_P1_ (398 aa). The comparison was generated by POLYVIEW2D. α-helices and β-sheets are displayed in red and green, respectively. Domains and motifs for ParA_F_ are indicated as follows: dimerization (1–27 aa), wHTH (winged helix turn helix; 40–100 aa), WA (Walker A; 114–122 aa), WA′ (Walker A′; 139–149 aa), WB (Walker B; 246–251 aa), nsDNA binding K340 (lysine 340). Vertical bars represent Hinge positions predicted by HingeProt analysis. (**B**) Model of ParA_F_ 3D structure generated by Swiss-Model using ADP-ParA_P1_ structure as a reference (see Materials and Methods). The two monomers are colored separately in grey and blue. For the right monomer, the first α-helix and the wHTH are displayed in yellow and red, respectively, and the Nter (N) and Cter (C) are indicated. Hinge positions are represented by black spheres. Each monomer contains an ADP represented with spheres, red for oxygen (O), green for carbon (C), blue for nitrogen (N) and orange for phosphate (P). (**C**) The wHTH of ParA_F_ undergoes a large conformational change. The B-factor, estimating the atomic fluctuation and expressed in angstrom (A^2^), was calculated from residues 5 to 381 from 40 ns Molecular Dynamic trajectories. The local mobility within the molecule is mostly restricted to the region encompassing the wHTH motif.

**Figure 5 genes-12-01345-f005:**
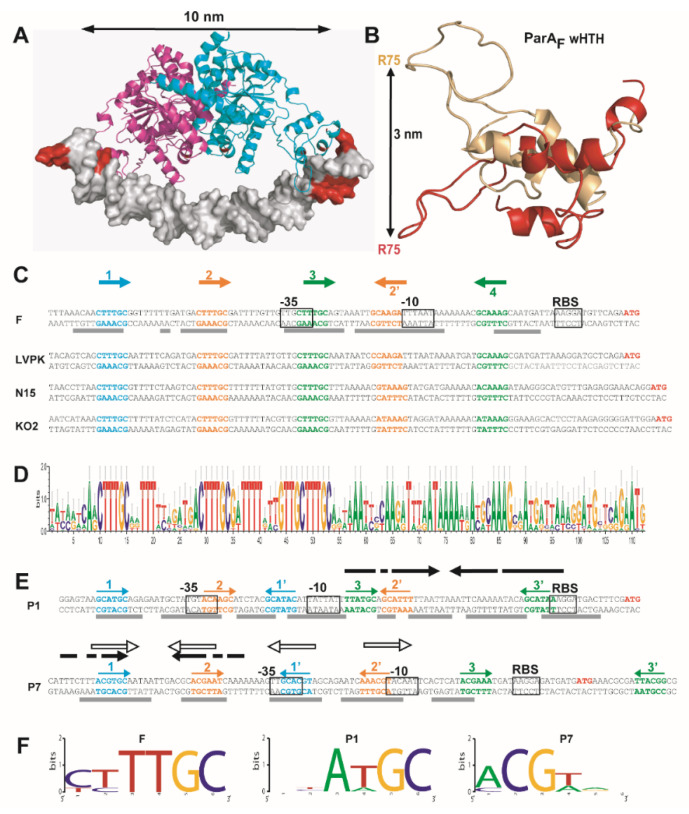
Type Ia plasmid ParA binds to overlapping motifs arranged as inverted repeats. (**A**) Docking ParA_F_ onto *P*parAB_F_. ParA_F_ from the modeled structure was docked onto a 40-bp DNA carrying the IR 3-4 motif. Different DNA bending angles (see Material and Methods) were assayed with one possible conformation of ParA_F_ dimer. The best docking on the binding motifs was observed with a 120° DNA curvature. ParA_F_ monomers are represented in magenta and cyan. The DNA is displayed in gray with the binding motifs #3 and #4 colored in red. (**B**) The winged-HTH domain of ParA_F_ undergoes large conformational rearrangement. The backbone of the wHTH is superimposed before (red) and after (salmon) 40 ns MD simulation. The distance between the C-α atoms of arginine 75 (R75) in the initial and final states is shown by a black line. The full structures are presented in Appendix A. (**C**) Par promoter regions of the plasmid F and close homologs display a strong conservation of the ParA binding motifs. The sequences of *P*parAB_F_ and of three close homologs from LVKP (large virulence plasmid pLVPK of *Klebsiella pneumoniae* CG43), N15 (linear prophage N15 of *E. coli*) and KO2 (phage phiKO2 of *Klebsiella oxytoca*) are shown up to the *parA* start codon (ATG) colored in red. The −10 and −35 transcriptional signals and the ribosome binding sites (RBS) are indicated by open boxes. Binding motifs #1 and IR 2-2′ and 3-4, are written in blue, orange and green, respectively, and are represented with corresponding colored arrows. The grey lines below the *P*parAB_F_ sequence corresponds to the regions protected by ParA_F_ in footprint experiments [21]. (**D**) Partition promoters related to *P*parAB_F_ display a high level of conservation for the inverted repeats motifs and their spacing. Sixteen different partition promoters recovered from BLASTn searches using the Par promoter regions of F, LVPK, N15 and KO2 were subjected to quantitative Logo analysis [35]. The sequences and the alignment of these promoter regions are presented in Appendix A. (**E**) Promoter regions of the *par* operons of plasmids P1 and P7. DNA sequences and promoter core signals are displayed with the same color codes and drawings as in C. Inverted black and open arrows indicate the imperfect repeat sequences and the DNA binding motifs, respectively, identified previously [10]. The regions protected from DNAse I footprinting by ParA_P1_ [32,36] and ParA_P7_ [10] are indicated by grey lines below each sequence. (**F**) Logo analysis of the proposed ParA binding sites for the plasmids F, P1 and P7. The six proposed binding motifs in the *P*parAB regions were subjected to Logo analysis, except for the plasmid F for which only five motifs were identified (see main text).

**Figure 6 genes-12-01345-f006:**
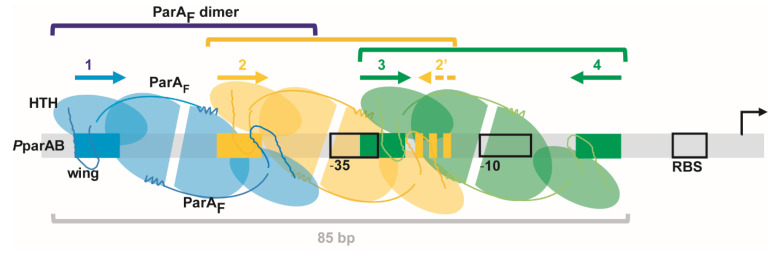
Model of ParA cooperative binding to plasmid partition promoters.

**Table 1 genes-12-01345-t001:** Measurement of ParA_F_ melting temperature. The melting temperature (Tm) of ParA_F_ (3 µM) was determined using a label-free fluorescence assay in presence or absence of ADP (1 mM) and/or 120-bp *P*parAB_F_ DNA (250 nM). The standard deviation (+/−) was calculated from triplicate measurements. The data shown are from a typical experiment. Each assay has been reproduced three times with the same variations observed between each tested condition.

	Tm (°C)
ParA_F_	44.5 +/− 0.4
ParA_F_ + *P*parAB_F_	45.1 +/− 0.2
ParA_F_ + ADP	46.1 +/− 0.1
ParA_F_ + *P*parAB_F_ + ADP	45.8 +/− 0.1

## Data Availability

The structure of the 3D model of ParA_F_ was deposited in the Protein Model DataBase (PMDB) with the ID number PM0084128.

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
