# Peer review of "Three ParA Dimers Cooperatively Assemble on Type Ia Partition Promoters"

_genes, 2021, doi:10.3390/genes12091345_

Round 1
Reviewer 1 Report
Within this paper the authors biochemically characterise the interaction between the ParA protein and its DNA binding site. Within the paper the author shows the binding of three dimers of the plasmid encoded ParA to its own promoter region showing how it is regulated. Overall the results are presented logically and the hypotheses tested rigorously with multiple complimentary methods.
The specific comments for review are below:
Line 77: A space is needed between the full stop and Bottom
Line 543: The authors have performed a series of ADP binding experiments, however in the previous papers cited the primary focus was on ATP binding and hydrolysis. Did the authors look at ATP binding and hydrolysis in this paper? It is also worth considering other nucleotides as there have been two recent papers (https://elifesciences.org/articles/53515, https://elifesciences.org/articles/67554) that have shown Par proteins binding to CTP.
Figures 2 and S1: Please add a colour key to the sensograms as it is difficult to work out which line corresponds to which concentration, although I understand why the authors have done it this way a colour line next to each would make this much easier for the reader.
I would also urge caution to the authors with regards to their use of homology models to make structural claims, whilst these are generally good predictors of structure they are never entirely accurate. For example in lines 368-370 the authors talk about "structural similarities" when this should be referred to as "predicted structural similarities".
Reviewer 2 Report
This is a straightforward, well-written paper with conclusions that are well-supported through multiple lines of independent biochemical evidence. I don't have any suggestions to further improve the paper. Aside from this, I don't have any suggestions that can further improve this paper.
Author Response
We are pleased that this reviewer comments very positively our study.